# An autoencoder and artificial neural network-based method to estimate parity status of wild mosquitoes from near-infrared spectra

**Masabho P. Milali**[1,2]\*, **Samson S. Kiware**[1,2], **Nicodem J. Govella**[2], **Fredros Okumu**[2], **Naveen Bansal**[1], **Serdar Bozdag**[3], **Jacques D. Charlwood**[4], **Marta F. Maia**[5,6,7], **Sheila B. Ogoma**[8], **Floyd E. Dowell**[9], **George F. Corliss**[1,10], **Maggy T. Sikulu-Lord**[11☯], **Richard J. Povinelli**[1,10☯]

1 Department of Mathematical and Statistical Sciences, Marquette University, Milwaukee, WI, United States of America, 2 Environmental Health and Ecological Sciences Thematic Group, Ifakara Health Institute, Ifakara, Tanzania, 3 Department of Computer Science, Marquette University, Milwaukee, WI, United States of America, 4 Liverpool School of Tropical Medicine, Liverpool, England, United Kingdom, 5 Wellcome Trust Research Programme, Kenya Medical Research Institute, Kilifi, Kenya, 6 Swiss Tropical and Public Health Institute, Basel, Switzerland, 7 University of Basel, Basel, Switzerland, 8 Clinton Health Access Initiative, Nairobi, Kenya, 9 Center for Grain and Animal Health Research, USDA, Agricultural Research Service, Manhattan, KS, United States of America, 10 Department of Electrical and Computer Engineering, Marquette University, Milwaukee, WI, United States of America, 11 The School of Public Health, University of Queensland, Brisbane, Queensland, Australia

☯ These authors contributed equally to this work.

\* pmasabho@ihi.or.tz

**Data Availability Statement:** All relevant data are within the manuscript and its Supporting Information files.

## Abstract

After mating, female mosquitoes need animal blood to develop their eggs. In the process of acquiring blood, they may acquire pathogens, which may cause different diseases in humans such as malaria, zika, dengue, and chikungunya. Therefore, knowing the parity status of mosquitoes is useful in control and evaluation of infectious diseases transmitted by mosquitoes, where parous mosquitoes are assumed to be potentially infectious. Ovary dissections, which are currently used to determine the parity status of mosquitoes, are very tedious and limited to few experts. An alternative to ovary dissections is near-infrared spectroscopy (NIRS), which can estimate the age in days and the infectious state of laboratory and semi-field reared mosquitoes with accuracies between 80 and 99%. No study has tested the accuracy of NIRS for estimating the parity status of wild mosquitoes. In this study, we train an artificial neural network (ANN) models on NIR spectra to estimate the parity status of wild mosquitoes. We use four different datasets: *An. arabiensis* collected from Minepa, Tanzania (Minepa-ARA); *An. gambiae s.s* collected from Muleba, Tanzania (Muleba-GA); *An. gambiae s.s* collected from Burkina Faso (Burkina-GA); and *An.gambiae s.s* from Muleba and Burkina Faso combined (Muleba-Burkina-GA). We train ANN models on datasets with spectra preprocessed according to previous protocols. We then use auto-encoders to reduce the spectra feature dimensions from 1851 to 10 and re-train the ANN models. Before the autoencoder was applied, ANN models estimated parity status of mosquitoes in Minepa-ARA, Muleba-GA, Burkina-GA and Muleba-Burkina-GA with out-of-sample accuracies of 81.9±2.8 (N = 274), 68.7±4.8 (N = 43), 80.3±2.0 (N = 48), and 75.7±2.5 (N = 91), respectively. With the autoencoder, ANN models tested on out-of-sample data

**Funding:** Data collection was funded by: Grand Challenges Canada Stars for Global Health funded by the government of Canada grant 043901 awarded to MTSL; DFID/MRC/Wellcome Trust through the Joint Health Trials Scheme awarded to JDC (Award Number MR/L004437/1); National Institute of Allergy and Infectious Diseases grant R01-AI094349-01A1; and Marquette University Graduate School, for studentship awarded to MPM.

**Competing interests:** The authors have declared that no competing interests exist.

achieved 97.1±2.2% (N = 274), 89.8 ± 1.7% (N = 43), 93.3±1.2% (N = 48), and 92.7±1.8% (N = 91) accuracies for Minepa-ARA, Muleba-GA, Burkina-GA, and Muleba-Burkina-GA, respectively. These results show that a combination of an autoencoder and an ANN trained on NIR spectra to estimate the parity status of wild mosquitoes yields models that can be used as an alternative tool to estimate parity status of wild mosquitoes, especially since NIRS is a high-throughput, reagent-free, and simple-to-use technique compared to ovary dissections.

## Introduction

Evaluation of existing malaria control interventions such as insecticide-treated nets (ITNs) and indoor residual spraying (IRS) relies upon, among other factors, the assessment of the changes occurring in the mosquito parity structure prior to and after implementation of an intervention [1–3]. The parity status of mosquitoes corresponds with their capability to transmit *Plasmodium* parasites, with an assumption that parous mosquitoes are more highly capable than nulliparous mosquitoes, as they may have accessed parasite-infected blood. A shift in the parity structure towards a population with more nulliparous mosquitoes signifies a reduction in the risk of disease transmission [2, 4, 5], as the chances that mosquitoes carry the malaria parasite declines [6].

The current standard technique for estimating the parity status of female mosquitoes involves dissection of their ovaries to separate mosquitoes into those that have previously laid eggs, known as the parous group (assumed to be old and potentially infectious), and those that do not have a gonotrophic history, known as the nulliparous group (assumed to be young and non-infectious) [7]. Another standard technique also based on the dissection of ovaries determines the number of times a female mosquito has laid eggs [8]. However, both techniques are laborious, time consuming, and require skilled technicians. These technical difficulties lead to analysis of small sample sizes that often fail to capture the heterogeneity of a mosquito population.

Near infrared spectroscopy (NIRS) complimented by techniques from machine learning, have been demonstrated to be alternative tools for predicting age, species, and infectious status of laboratory and semi-field raised mosquitoes [9–20]. NIRS is a rapid, non-invasive, reagent-free technique that requires minimal skills to operate, allowing hundreds of samples to be analyzed in a day. However, the accuracy of NIRS techniques for predicting the parity status of wild mosquitoes has not been tested. Moreover, recently, it has been reported that models trained on NIR spectra using an artificial neural network (ANN) estimate the age of laboratory-reared *An. arabiensis*, *An.gambiae*, *Aedes aegypti*, and *Aedes albopictus* with accuracies higher than models trained on NIR spectra using partial least squares (PLS) [20].

In this study, we train ANN models on NIR spectra preprocessed according to an existing protocol [9] to estimate the parity status of wild *An. gambiae* s.s. and *An. arabiensis*. We then apply autoencoders to reduce the spectra feature space from 1851 to 10 and re-train ANN models. The ANN model achieved an average accuracy of 72% and 93% before and after applying the autoencoder, respectively. These results suggest ANN models trained on autoencoded NIR spectra as an alternative tool to estimate the parity status of wild *An. gambiae* and *An. arabiensis*. High-throughput, non-invasive, reagent free, and simple to use NIRS analyses compliment the limitations of ovary dissections.

## Materials and methods

### Ethics approvals

Ethics approvals for collecting mosquitoes in Minepa-ARA, Burkina-GA and Muleba-GA datasets from residents' homes were obtained from Ethics Review Boards of the Ifakara Health Institute (IHI-IRB/No. 17–2015), the Colorado State University (approval No. 09-1148H), and the Kilimanjaro Christian Medical College (Certificate No. 781), respectively.

### Data

We use data from wild *An. arabiensis* (Minepa-ARA) collected from Minepa, a village in southeastern Tanzania (published in [21] and publicly available for reuse), from wild *An. gambiae* s.s (Muleba-GA) collected from Muleba, northwestern Tanzania (mosquitoes published in [22], permission to reuse was obtained from the senior author) and from wild *An. gambiae* s.s collected from Bougouriba and Diarkadou-gou villages in Burkina Faso (Burkina-GA) (published in [12] and publicly available for reuse).

Mosquitoes in the Minepa-ARA and Muleba-GA datasets were captured using CDC light traps placed inside residential homes. Mosquitoes that were morphologically identified as members of the *Anopheles gambiae* complex were processed further. Prior to scanning, wild mosquitoes collected in Minepa were killed by freezing them for 20 minutes in a freezer that is calibrated to -20º C. After freezing the mosquitoes were re-equilibrated to room temperature for 30 minutes. Wild mosquitoes collected in Muleba were killed using 75% ethanol, dissected according to the technique described by Detinova [23] to determine their parity status, and preserved in silica gel. Mosquitoes in Minepa-ARA were dissected after scanning. Following a previous published protocol to collect spectra [9], mosquitoes in both Minepa-ARA and Muleba-GA were scanned using a LabSpec 5000 near-infrared spectrometer with an integrated light source (ASD Inc., Malvern, UK). After spectra collection, mosquitoes in Minepa-ARA were dissected to score their parity status. Then polymerase chain reaction (PCR) was conducted on DNA extracted from mosquito legs (in both Minepa-ARA and Muleba-GA) to identify species type as previously described [24]. Each mosquito was labeled with a unique identifier code linking each NIR spectrum to parity dissection and PCR information.

Data from wild *An. gambiae* s.s from Burkina Faso were published in [12] and publicly available for reuse. These mosquitoes are referred to as independent test sets 2 and 3 (ITS 2 and ITS 3) in [12]. ITS 2 has 40 nulliparous and 40 parous mosquitoes, and ITS 3 has 40 nulliparous and 38 parous mosquitoes. In this study, we combine these two datasets into one dataset and refer it as Burkina-GA. Mosquitoes in Burkina-GA (N = 158) were collected in 2013 in Burkina Faso from Bougouriba and Diarkadou-gou villages using either indoor aspiration or a human baited tent trap, and their ovaries were dissected according to the Detinova method [23]. Mosquitoes were preserved in silica gel before their spectra were collected using a LabSpec4i spectrometer (ASD Inc., Boulder, CO, USA).

### Model training and testing

We trained models on four datasets, namely Minepa-ARA, Muleba-GA, Burkina-GA, and Muleba-Burkina-GA (Muleba-GA and Burkina-GA combined). Before training models, spectra in all datasets were pre-processed according to the previously published protocol [9] and divided into two groups (nulliparous and parous). Spectra in the nulliparous and parous groups were labeled zero and one, respectively. The two groups were then merged, randomized, and divided into a training set (75%; N = 927 for Minepa-ARA, N = 140 for the Muleba-GA, N = 158 for Burkina-GA and N = 298 for Muleba-Burkina-GA) and a test set (the

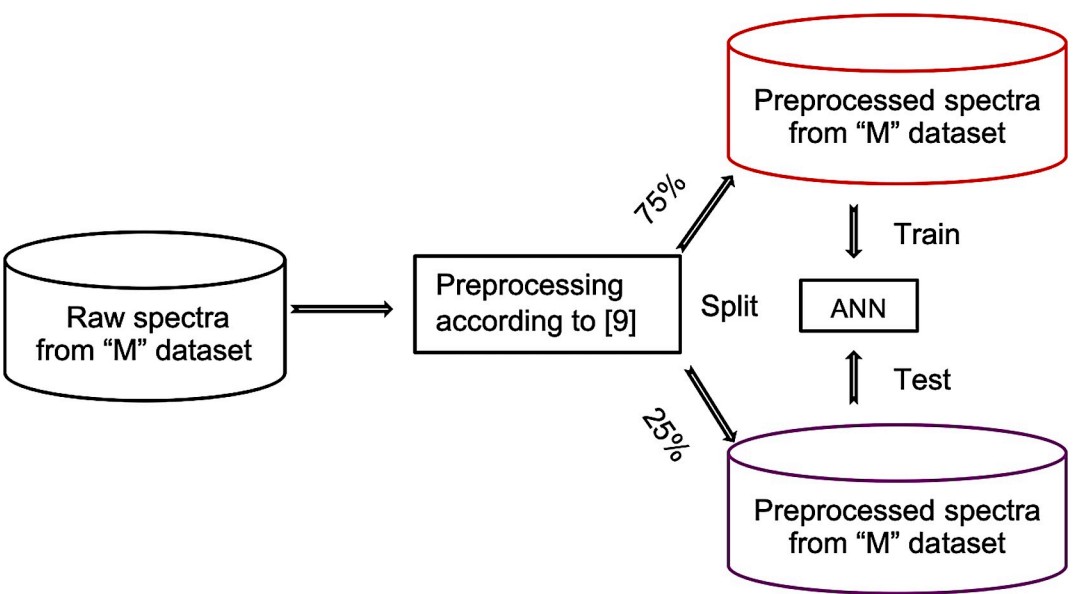

**Fig 1. Training and testing ANN model on spectra preprocessed according to Mayagaya et al. [9].** "M" is either Minepa-ARA, Muleba-GA, Burkina-GA, or Muleba-Burkina-GA.

remaining 25% in each dataset). On each dataset, using ten Monte-Carlo cross validations [20, 25] and Levenberg-Marquardt optimization, a one hidden layer, ten-neuron feed-forward ANN model with logistic regression as a transfer function was trained and tested in MATLAB (Fig 1).

Based on the accuracy of the model presented in Table 1 in the Results and Discussion section, we explored how to improve the model accuracy. Normally a parous class, unlike a nulliparous class, often is represented by a limited number of samples, posing a problem of data imbalance during model training. In this case, a large amount of data is required to obtain enough samples in a parous class for a model to learn and characterize it accurately. Obtaining enough data for model training is always challenging. The most common ways of dealing with the data imbalance are either to discard samples from a nulliparous class to equal the number of samples in a parous class or to bootstrap samples in a parous class [26]. However, discarding data to equalize the data distribution in two classes in the training set leaves an imbalanced test set. Also, it is this imbalanced scenario to which the model will be applied in real cases. In addition, throwing away samples, especially from data sets with a high dimension feature space, can lead to over-fitting the model. Alternatively, for datasets with a high dimension feature space, instead of discarding data from a class with a large number of samples, feature reduction techniques are employed [26]. Feature reduction reduces the size of the hypothesis space initially presented in the original data, thereby reducing the size of data required to adequately train the model. Principal component analysis (PCA) and partial least squares (PLS) are the commonly used unsupervised and supervised feature reduction methods, respectively,

**Table 1. Accuracies of reconstructing original feature spaces from encoded feature spaces.** MSE = mean square error.

| Metric | Steps | Encoded-Minepa-ARA | Encoded-Muleba-GA | Encoded-Burkina-GA |
|--------|-------|--------------------|--------------------|--------------------|
| MSE | Step 1 | 0.0046 | 0.0029 | 0.0031 |
| | Step 2 | 0.00005 | 0.0027 | 0.0022 |
| | Step 3 | 0.00008 | 0.0029 | 0.0011 |

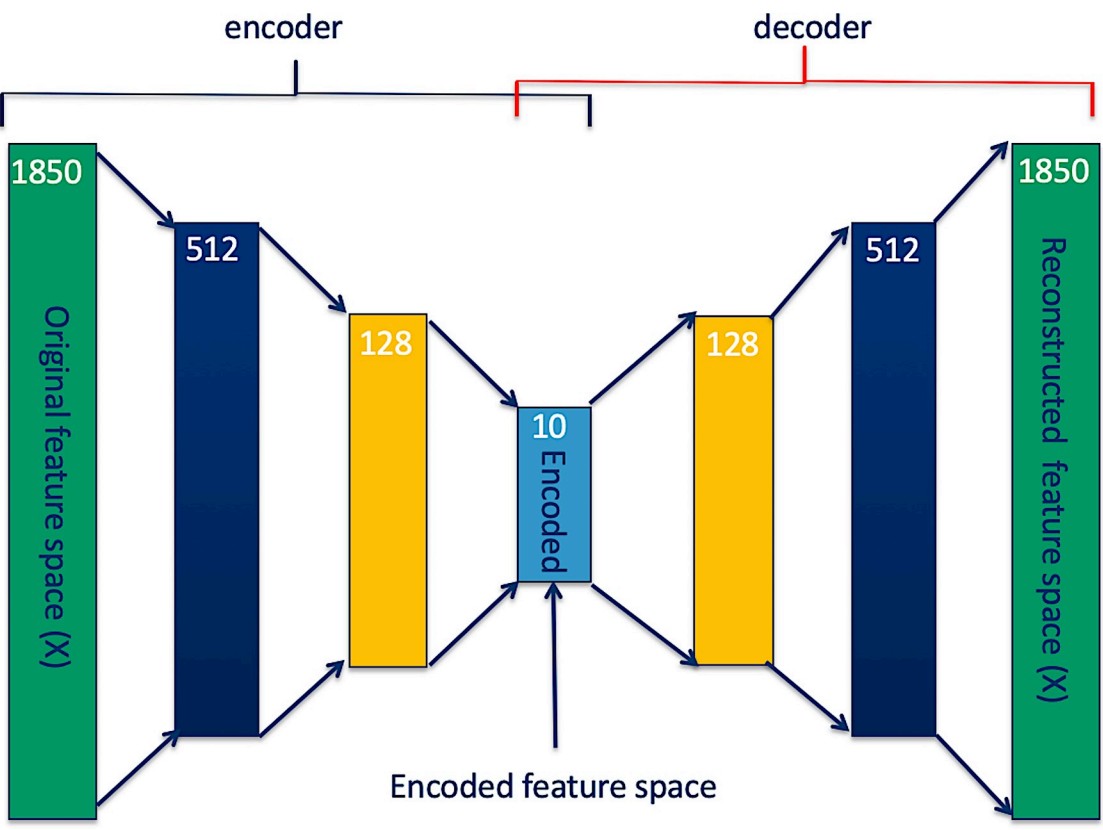

**Fig 2. Autoencoder reducing feature space dimension.**

especially for cases whose features are linearly related [27, 28]. Autoencoders recently are used as an alternative to PCA in cases involving both linear and non-linear relationships [29–32].

An autoencoder is an unsupervised ANN that learns both linear and non-linear relationships present in data and represents them in a new reduced dimension data space (which also can be used to regenerate the original data space) without losing important information [33–35]. The autoencoder has two parts, the encoder part, where an original dataset is encoded to a desired reduced feature space (encoded dataset) and the decoder part where the encoded dataset is decoded to an original dataset to determine how accurately the encoded dataset represents the original dataset. Fig 2 illustrates an example of an autoencoder in which an 1850-feature dataset is stepwise encoded to a 10-feature dataset. There is no formula for the number and size of steps to take to get to a desired feature size. However, taking several steps results on losing very little information, compared with taking a single step.

Once an encoded feature space can reconstruct the original feature space with an acceptable accuracy, the decoder is detached, and a desired model (in our case an ANN binary classifier) is trained on the encoded feature space as shown in Fig 3.

Egg laying appears to be affected by both linear and non-linear relationships. Hence, we separately train autoencoders on the Minepa-ARA, Muleba-GA, Burkina-GA, and Muleba-Burkina-GA datasets to reduce spectra feature dimensions from 1851 to 10 (Fig 4).

Table 1 presents accuracies of reconstructing original feature spaces from their respective encoded feature spaces. We refer to the autoencoded Minepa-ARA, Muleba-GA, Burkina-GA, and Muleba-Burkina-GA datasets as Encoded-Minepa-ARA, Encoded-Muleba-GA, Encoded-Burkina-GA, and Encoded-Muleba-Burkina-GA, respectively. We then train ANN models on

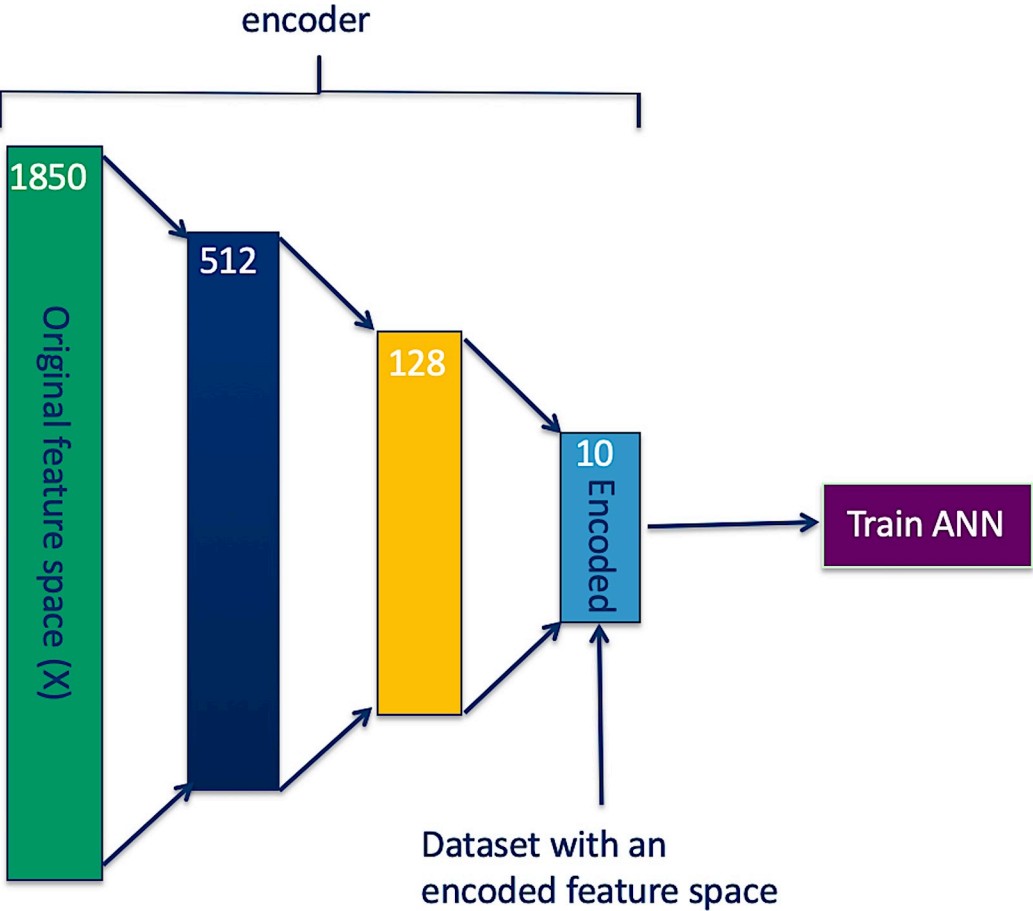

**Fig 3. ANN model trained on a dataset with an encoded feature space.**

Encoded-Minepa-ARA, Encoded-Muleba-GA, Encoded-Burkina-GA, and Encoded-Muleba-Burkina-GA (Fig 5).

Finally, we used the Encoded-Burkina-GA and the Encoded-Muleba-GA datasets as independent test sets to test accuracies of ANN models trained on the Encoded-Muleba-GA dataset and on the Encoded-Burkina-GA dataset, respectively (Fig 6A and 6B).

## Results and discussion

In this study, we demonstrated that near-infrared spectroscopy (NIRS) can estimate accurately the parity status of wild collected *An. arabiensis* and *An. gambiae* s.s. Referring to the published results in [11] (ANN models achieve higher accuracies than PLS models), we trained and tested an ANN model on NIRS spectra in four different datasets, pre-processed according to a previously published protocol [9]. The model achieved accuracies between 68.7 and 81.9% (Table 2, Figs 7 and 8). Table 2 further presents various metrics to score the performance of our classifiers, namely sensitivity, specificity, precision, and area under the receiver operating characteristic (ROC) curve (AUC). We calculated sensitivity, specificity, precision and accuracy of the model using Equs 1, 2, 3, and 4, respectively [36–39]. Sensitivity (also known as recall) is the percentage of correctly predicted parous mosquitoes, specificity is the percentage of correctly predicted nulliparous mosquitoes [20], and precision is the proportion of true parous mosquitoes out of all mosquitoes estimated by the model as parous [39]. We presented

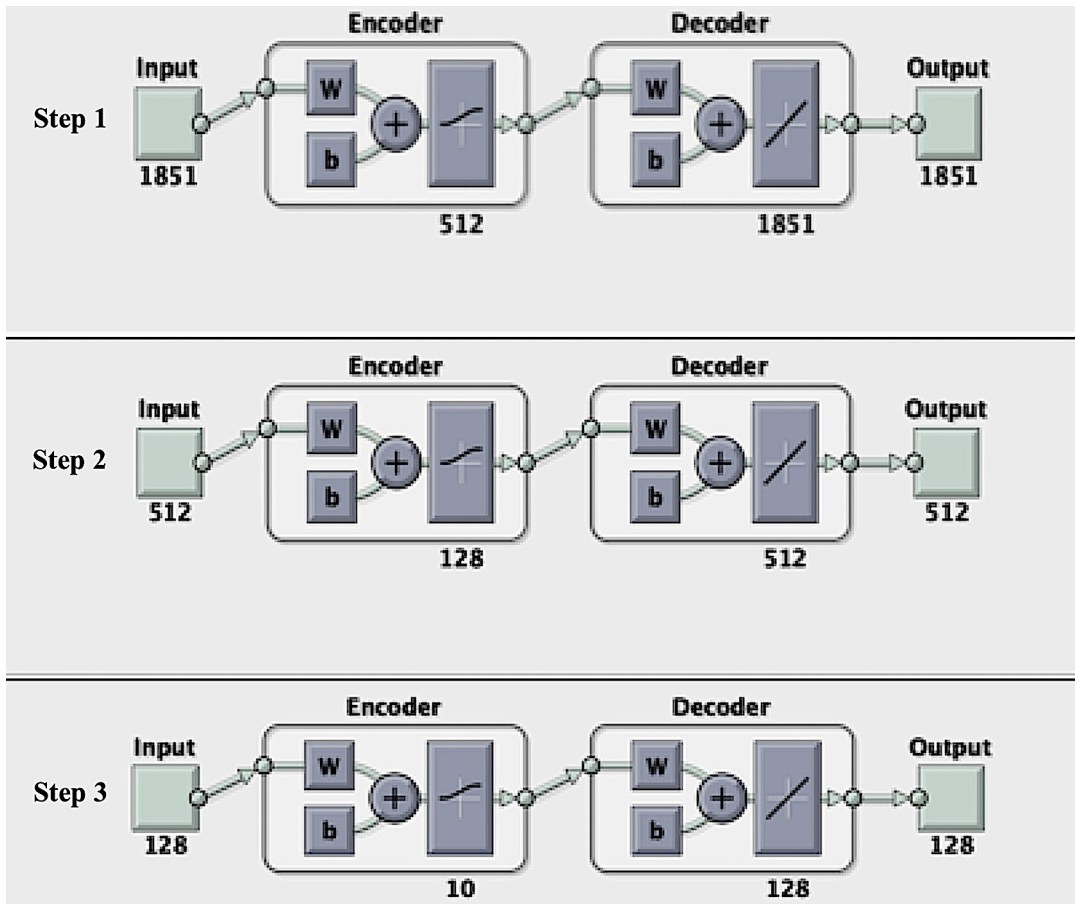

**Fig 4. Reducing spectra feature space using an autoencoder and re-constructing original feature spaces from their respective encoded feature spaces (reconstruction accuracies presented in Table 1).** Figures generated from MATLAB.

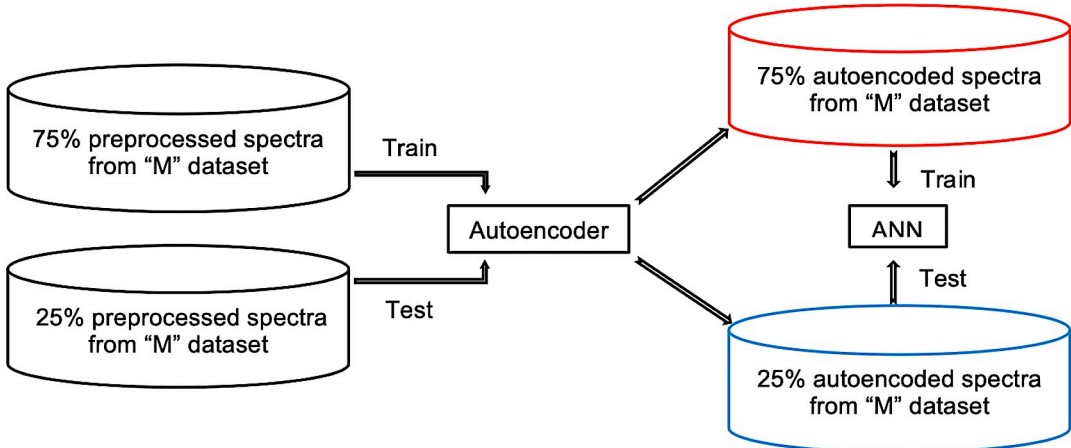

**Fig 5. Training and testing of ANN model on autoencoded spectra.** M is either Minepa-ARA, Muleba-GA, Burkina-GA, or Muleba-Burkina-GA dataset.

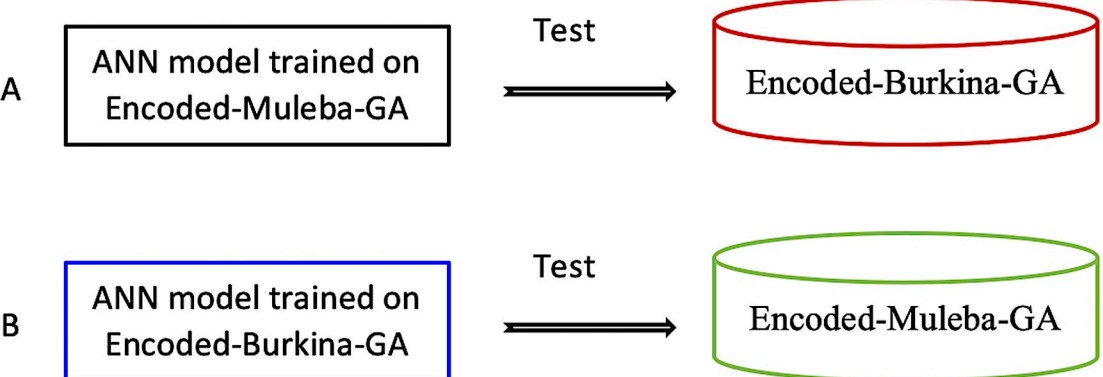

**Fig 6. Independent testing of ANN model trained on encoded datasets.** A) Applying an ANN model trained on the Encoded-Muleba-GA dataset to estimate the parity status of mosquitoes in the autoencoded Burkina-GA dataset. B) Applying the ANN model trained on the Encoded-Burkina-GA dataset to estimate the parity status of mosquitoes in the Encoded-Muleba-GA dataset.

both sensitivity and precision because different scholars prefer one metric to another especially for cases with imbalanced data [39].

Let

True Positive (TP) = Number of mosquitoes correctly classified by the model as parous,

False Positive (FP) = Number of mosquitoes wrongly classified by the model as parous,

True Negative (TN) = Number of mosquitoes correctly classified by the model as nulliparous,

Positive (P) = Total number of mosquitoes in test set that are parous, and

Negative (N) = Total number of mosquitoes in test set that are nulliparous.

Then

$$\text{Sensitivity} = \frac{\text{TP}}{\text{P}}, \tag{1}$$

$$\text{Specificity} = \frac{\text{TN}}{\text{N}}, \tag{2}$$

$$\text{Accuracy} = \frac{\text{TP} + \text{TN}}{\text{P} + \text{N}}, \text{ and} \tag{3}$$

**Table 2. Performance of an ANN model trained on 75% of mosquito spectra with 1851 features (before autoencoder) and tested on the remaining 25% spectra (out of the sample testing).** AUC values are the area of the ROC curves in Fig 8. Minepa-ARA (Nulliparous = 656, Parous = 271), Muleba-GA (Nulliparous = 119, Parous = 21) Burkina-GA (Nulliparous = 80, Parous = 78).

| | Minepa-ARA (N = 927) | Muleba-GA (N = 140) | Burkina-GA (N = 158) | Muleba-Burkina-GA (N = 298) |
|---|---|---|---|---|
| **Accuracy (%)** | 81.9 ± 2.8 | 68.7 ± 4.8 | 80.3 ± 2.0 | 75.7 ± 2.5 |
| **Sensitivity (%)** | 79.7 ± 3.2 | 37.8 ± 6.6 | 76.5 ± 2.1 | 70.2 ± 3.1 |
| **Specificity (%)** | 86.0 ± 1.6 | 80.1 ± 2.7 | 88.3 ± 2.3 | 77.6 ± 2.9 |
| **Precision (%)** | 74.3 ± 3.4 | 31.3 ± 5.2 | 77.8 ± 1.8 | 68.8 ± 3.2 |
| **AUC (%)** | 77.2 | 55.9 | 83.6 | 76.4 |

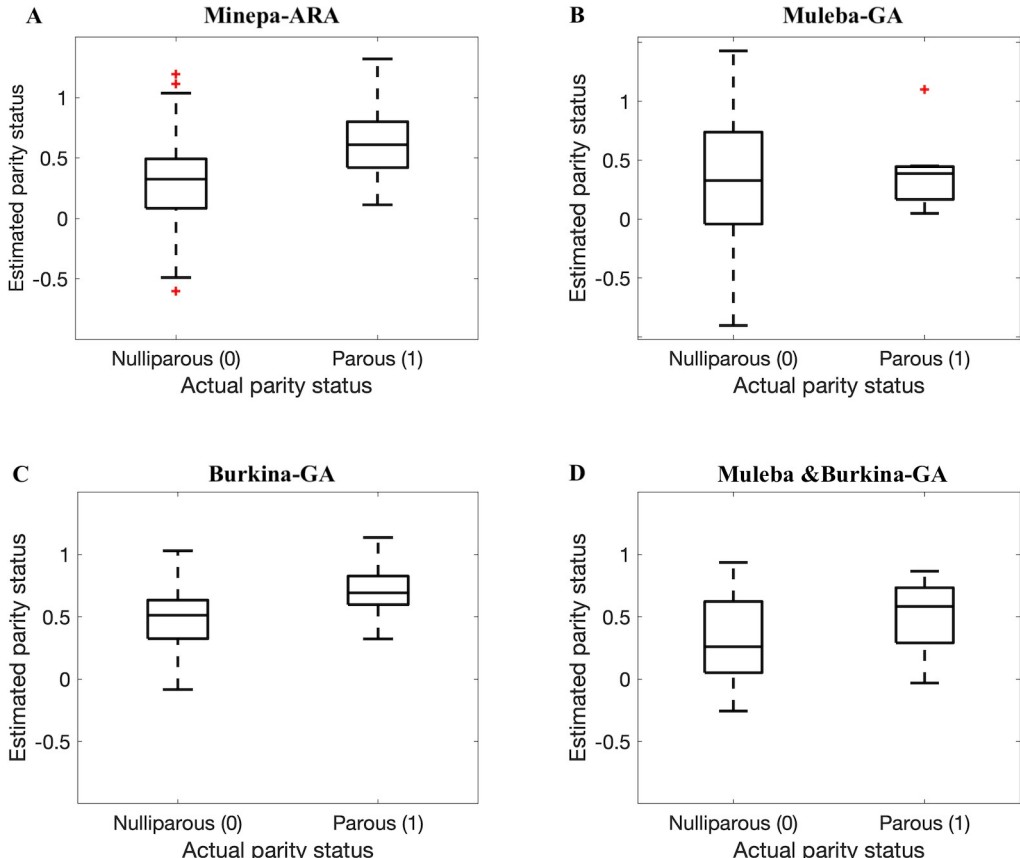

**Fig 7. Box plots of parity estimation score when ANN models trained on 75% of spectra before the autoencoder was applied and tested on the remaining spectra (25%) (out of the sample testing).** A, B, C, and D represent results for the Minepa-ARA, Muleba-GA, Burkina-GA, and Muleba-Burkina-GA (mosquitoes in Muleba-GA and Burkina-GA datasets combined) datasets, respectively.

$$Precision = \frac{TP}{TP + FP}. \qquad (4)$$

AUC was computed from the receiver operating characteristic (ROC) curve shown in Fig 8 generated by plotting the true parous rate against the false parous rate at different threshold settings. A higher AUC is interpreted as higher predictivity performance of the model [40, 41]. The ROC curve normally presents the performance of the model at different thresholds (cut off points), providing more information on the accuracy of the classifier [40, 41]. Table 3 provides confusion matrices from the last (tenth) Monte-Carlo cross validation showing model accuracy in absolute values.

We hypothesized that results presented in Tables 2 and 3, and in Figs 7 and 8 were influenced by the size of a dataset used to train the model. The model that was trained on a dataset with a relatively larger number of mosquitoes, especially in the parous class, performed better than the model trained on the dataset with fewer mosquitoes.

The current standard preprocessing technique [9] leaves a mosquito spectrum with an 1851- dimensional feature space. Mathematically, binary inputs with a 1851-dimensional feature space present $2^{2^{(1851)}}$ hypothesis space dimensions for the model to learn [42–44].

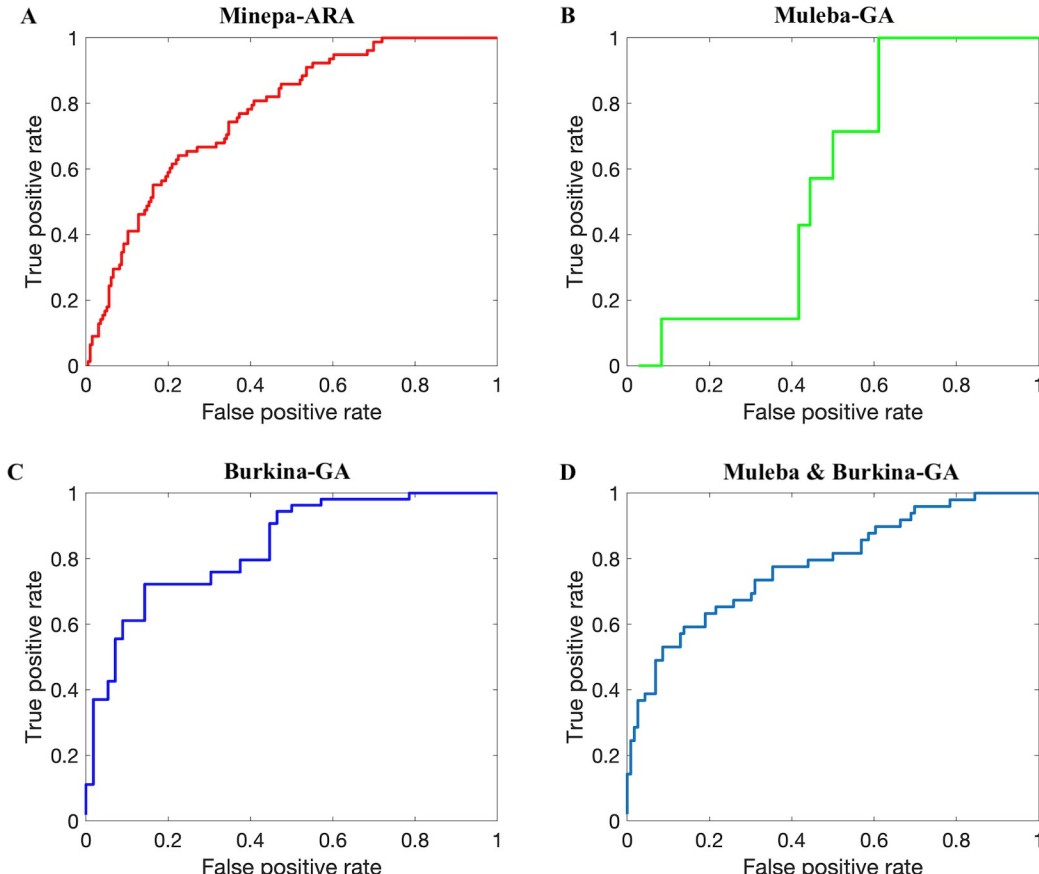

**Fig 8. ROC curves (AUCs presented in the last row of Table 2) showing results when ANN models trained on 75% of spectra before the autoencoder was applied and tested on the remaining spectra (25%) (out of the sample testing).** A, B, C, and D represent results for the Minepa-ARA, Muleba-GA, Burkina-GA, and Muleba-Burkina-GA (mosquitoes in Muleba-GA and Burkina-GA datasets combined) datasets, respectively. In each ROC curve, a threshold of 0.5 was used to compute true positive rate and false positive rate.

**Table 3. Confusion matrices showing accuracies of the models in absolute values when the models were trained on spectra before feature reduction by autoencoder.** A) Minepa-ARA, B) Muleba-GA, C) Burkina-GA and D) Muleba-Burkina-GA. Results from the last Monte-Carlo cross validation.

| | | Actual Parity | | |
|---|---|---|---|---|
| | **Estimates** | **Nulliparous** | **Parous** | **Total** |
| | Nulliparous | 165 | 17 | 182 |
| A | Parous | 31 | 61 | 92 |
| | Total | 196 | 78 | 274 |
| | Nulliparous | 28 | 4 | 32 |
| B | Parous | 8 | 3 | 11 |
| | Total | 36 | 7 | 43 |
| | Nulliparous | 20 | 6 | 26 |
| C | Parous | 4 | 18 | 22 |
| | Total | 24 | 24 | 48 |
| | Nulliparous | 46 | 9 | 55 |
| D | Parous | 14 | 22 | 36 |
| | Total | 60 | 31 | 91 |

**Table 4. Performance of an ANN model trained on 75% of the encoded mosquito spectra (10 features) and tested on the remaining 25% of the encoded mosquito spectra.** AUC values are the area of the ROC curves in Fig 10. Minepa-ARA (Nulliparous = 656, Parous = 271), Muleba-GA (Nulliparous = 119, Parous = 21), Burkina-GA (Nulliparous = 80, Parous = 78).

| | Minepa-ARA (N = 927) | Muleba-GA (N = 140) | Burkina-GA (N = 158) | Muleba-Burkina-GA (N = 298) |
|---|---|---|---|---|
| **Accuracy (%)** | 97.1 ± 2.2 | 89.8 ± 1.7 | 93.3 ± 1.2 | 92.7 ± 1.8 |
| **Sensitivity (%)** | 94.9 ± 1.6 | 70.1 ± 2.3 | 91.7 ± 1.9 | 88.2 ± 2.9 |
| **Specificity (%)** | 98.6 ± 1.3 | 96.9 ± 1.2 | 96.4 ± 1.6 | 94.7 ± 2.1 |
| **Precision (%)** | 93.7 ± 2.4 | 62.5 ± 3.2 | 91.3 ± 1.4 | 93.1 ± 2.5 |
| **AUC (%)** | 96.7 | 91.5 | 93.1 | 94.9 |

Successful learning of such hypothesis space dimensions requires many data points (mosquitoes in our case). Finding enough wild mosquitoes, especially parous mosquitoes, for a model to learn such a hypothesis space is expensive and time consuming. Feature reduction is an alternative to overcome this, as it reduces the hypothesis space dimension initially presented by the original data, hence lowering the number of data required to train the model efficiently. Techniques such as principal component analysis (PCA) [27, 28], partial least squares (PLS) [27, 45, 46], singular value decomposition (SVD) [30, 46, 47], and autoencoders can reduce a feature space to a size that can be learned by the available data without losing important information. PCA, PLS, and SVD are commonly used when features are linearly dependent [27, 28], otherwise, an autoencoder, which can be thought as a nonlinear version of PCA, is used [29–32].

Therefore, we applied an autoencoder as illustrated in Figs 2 and 4 to reduce the spectra feature space from 1851 features to 10 features (Table 1 presents the accuracies of reconstructing original feature spaces from the encoded (reduced) feature spaces), cutting down the hypothesis space dimensions from $2^{2^{(1851)}}$ to $2^{2^{(10)}}$, and re-trained ANN models (Figs 3 and 5). As presented in Tables 4 and 5, and in Figs 9 and 10, the accuracy of the model improved from an average of 72% to 93%, suggesting an ANN model trained on autoencoded NIR spectra as an appropriate tool to estimate the parity status of wild mosquitoes.

We further applied a model trained on encoded Muleba-GA dataset to estimate the parity status of mosquitoes in the encoded Burkina-GA dataset and a model trained on encoded Burkina-GA dataset to estimate the parity status of mosquitoes in encoded Muleba-GA. Here we

**Table 5. Confusion matrices showing accuracies of the models in absolute values when the models were trained on spectra after feature reduction by autoencoder.** A) Minepa-ARA, B) Muleba-GA, C) Burkina-GA, and D) Muleba-Burkina-GA. Results from the last Monte-Carlo cross validation.

| | | Actual Parity | | |
|---|---|---|---|---|
| | **Estimates** | **Nulliparous** | **Parous** | **Total** |
| | Nulliparous | 192 | 7 | 199 |
| A | Parous | 4 | 71 | 95 |
| | Total | 196 | 78 | 274 |
| | Nulliparous | 33 | 2 | 35 |
| B | Parous | 3 | 5 | 8 |
| | Total | 36 | 7 | 43 |
| | Nulliparous | 22 | 3 | 25 |
| C | Parous | 2 | 21 | 23 |
| | Total | 24 | 24 | 48 |
| | Nulliparous | 58 | 4 | 62 |
| D | Parous | 2 | 27 | 29 |
| | Total | 60 | 31 | 91 |

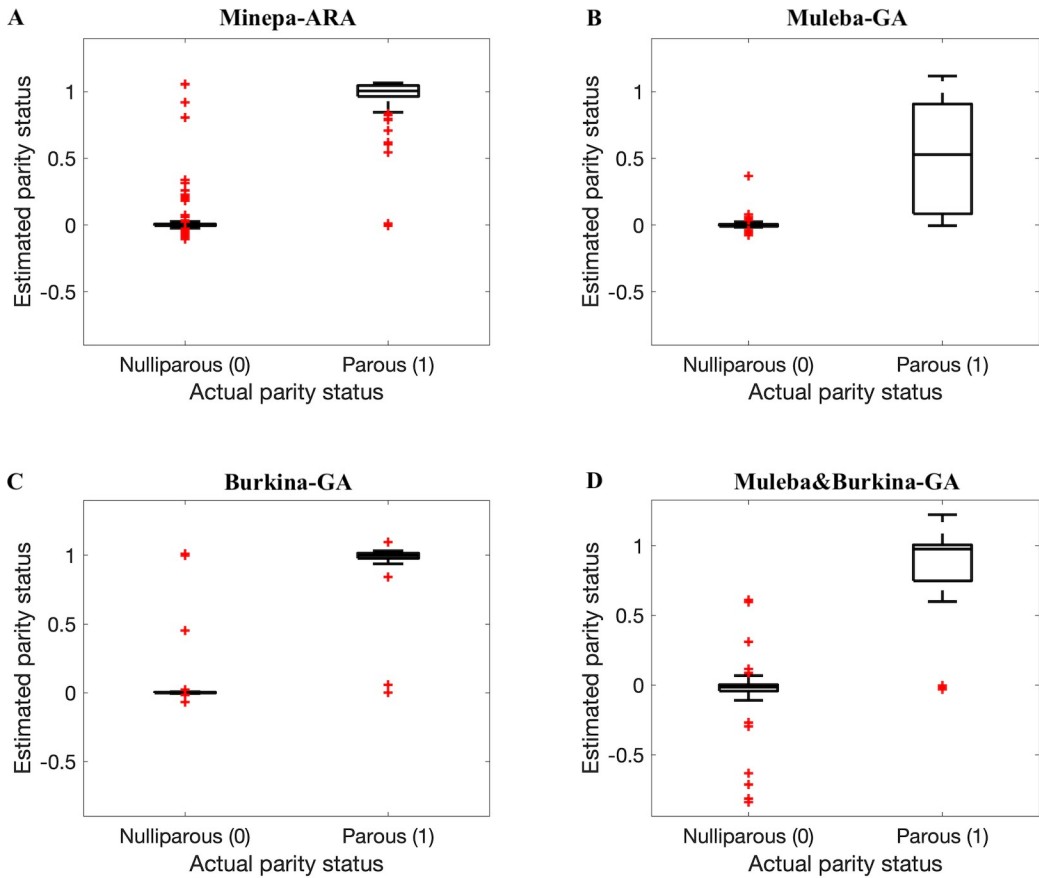

**Fig 9. Box plots showing results when ANN models trained on 75% of encoded spectra in datasets were tested on the remaining encoded spectra (25%).** A, B, C, and D represent results for the Encoded-Minepa-ARA, Encoded-Muleba-GA, Encoded-Burkina-GA, and Encoded-Muleba & Burkina-GA (mosquitoes in Encoded-Muleba-GA and Encoded-Burkina-GA datasets combined) datasets, respectively.

wanted to test how the model performs on mosquitoes from different cohorts. As presented in Table 6, the model performed with accuracies of 68.6% and 88.3%, respectively, showing a model trained on encoded Burkina-GA dataset extrapolates well to mosquitoes from a different cohort than a model trained on the encoded Muleba-GA dataset.

A possible explanation of the results shown in Table 6 could be that, unlike for the Burkina-GA dataset, the number of parous mosquitoes (N = 21) in the Muleba-GA dataset was not representative enough for a model to learn important characteristics that extrapolate to mosquitoes in a cohort other than the one used to train the model. Although the Muleba-GA model had the poor sensitivity as presented in Table 6, the Burkina-GA model results still suggest that ANN model trained on acceptable number of both encoded parous and nulliparous can be applied to estimate parity status of mosquitoes from different cohorts other than the one used to train the model.

## Conclusion

These results suggest that applying autoencoders and artificial neural networks to NIRS spectra as an appropriate complementary method to ovary dissections to estimate parity status of wild mosquitoes. The high-throughput nature of near-infrared spectroscopy provides a statistically acceptable sample size to draw conclusions on parity status of a particular wild mosquito

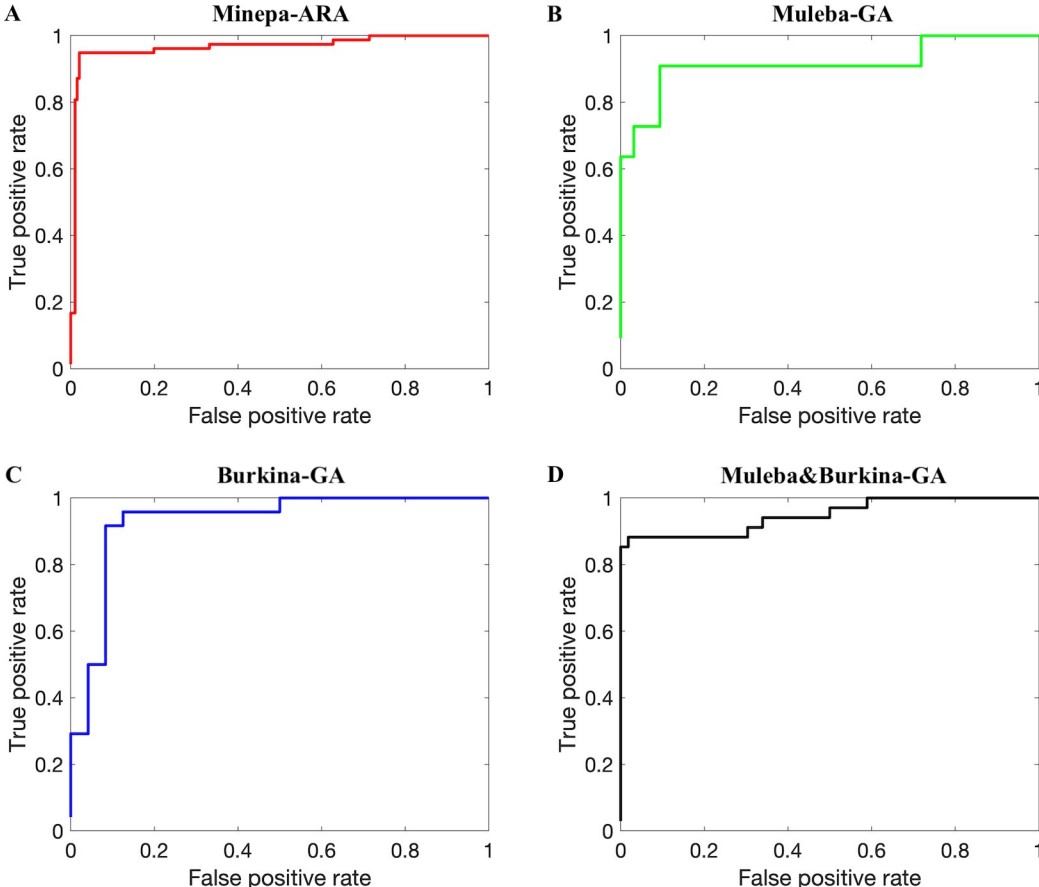

**Fig 10. ROC curves (AUCs presented in the last row of Table 4) showing results when ANN models trained on 75% of encoded spectra were tested on the remaining encoded spectra (25%).** A, B, C, and D represent results for the Encoded-Minepa-ARA, Encoded-Muleba-GA, Encoded-Burkina-GA, and Encoded-Muleba-Burkina-GA (mosquitoes in Encoded-Muleba-GA and Encoded-Burkina-GA datasets combined) datasets, respectively. In all ROC curves, a 0.5 threshold was used to calculate false positive rate and true positive rate.

population. Before this method can be used as a stand-alone method to estimate the parity status of wild mosquitoes, we suggest repeating of the analysis on different datasets with much larger mosquito sample sizes to test the reproducibility of the results. Hence, with the results presented in this manuscript, we recommend complementing ovary dissection with ANN models trained on NIRS spectra with their feature reduced by an autoencoder to estimate parity status of wild mosquito population.

**Table 6. Independent testing of ANN models trained on Muleba-GA and Burkina-GA encoded datasets.**

|  | ANN model trained on Encoded-Muleba-GA, tested on Encoded-Burkina-GA | ANN model trained on Encoded-Burkina-GA, tested on Encoded-Muleba-GA |
|---|---|---|
| **Accuracy (%)** | 68.6 | 88.3 |
| **Sensitivity (%)** | 26.5 | 86.1 |
| **Specificity (%)** | 94.4 | 92.2 |

## Supporting information

**S1 Appendix. Zip folder with Minepa-ARA, Muleba-GA and Burkina-GA datasets.** Column header, wavelengths in 'nm'.
(ZIP)

## Acknowledgments

We extend our gratitude to Benjamin Krajacich for allowing us to use his already-published datasets (Burkina-GA dataset) in our analyses; the USDA, Agricultural Research Service, Center for Grain and Animal Health Research, USA for loaning us the near-infrared spectrometer used to scan the mosquitoes; and Gustav Mkandawile who worked tirelessly to make sure we obtained mosquitoes in the Minepa-ARA and Muleba-GA datasets.

Mention of trade names or commercial products in this publication is solely for the purpose of providing specific information and does not imply recommendation or endorsement by the U.S. Department of Agriculture. USDA is an equal opportunity provider and employer.

## Author Contributions

**Conceptualization:** Masabho P. Milali, Samson S. Kiware, George F. Corliss, Richard J. Povinelli.

**Data curation:** Masabho P. Milali, Samson S. Kiware, Nicodem J. Govella, Jacques D. Charlwood, Maggy T. Sikulu-Lord.

**Formal analysis:** Masabho P. Milali.

**Funding acquisition:** Masabho P. Milali, Fredros Okumu, Jacques D. Charlwood, Marta F. Maia, Sheila B. Ogoma, George F. Corliss, Maggy T. Sikulu-Lord, Richard J. Povinelli.

**Investigation:** Masabho P. Milali.

**Methodology:** Masabho P. Milali, Samson S. Kiware, George F. Corliss, Richard J. Povinelli.

**Project administration:** Masabho P. Milali, Fredros Okumu, Marta F. Maia, Maggy T. Sikulu-Lord.

**Resources:** Floyd E. Dowell, George F. Corliss, Richard J. Povinelli.

**Software:** Masabho P. Milali, Richard J. Povinelli.

**Supervision:** Samson S. Kiware, Fredros Okumu, Naveen Bansal, Serdar Bozdag, George F. Corliss, Maggy T. Sikulu-Lord, Richard J. Povinelli.

**Validation:** Masabho P. Milali, Samson S. Kiware, Nicodem J. Govella, Fredros Okumu, Naveen Bansal, Serdar Bozdag, Jacques D. Charlwood, Floyd E. Dowell, George F. Corliss, Maggy T. Sikulu-Lord, Richard J. Povinelli.

**Visualization:** Masabho P. Milali, Samson S. Kiware, Richard J. Povinelli.

**Writing – original draft:** Masabho P. Milali.

**Writing – review & editing:** Masabho P. Milali, Samson S. Kiware, Nicodem J. Govella, Fredros Okumu, Naveen Bansal, Serdar Bozdag, Jacques D. Charlwood, Marta F. Maia, Sheila B. Ogoma, Floyd E. Dowell, George F. Corliss, Maggy T. Sikulu-Lord, Richard J. Povinelli.

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
