## [Decision Letter · Decision Letter 0]

1 Apr 2020

PONE-D-20-02482

An Autoencoder and Artificial Neural Network-based Method to Estimate Parity Status of Wild Mosquitoes from Near-infrared Spectra

PLOS ONE

Dear Mr Milali,

Thank you for submitting your manuscript to PLOS ONE. After careful consideration, we feel that it has merit but does not fully meet PLOS ONE’s publication criteria as it currently stands. Therefore, we invite you to submit a revised version of the manuscript that addresses the points raised during the review process.

Please revise the paper by taking into account the reviewer's comments.

We would appreciate receiving your revised manuscript by May 16 2020 11:59PM. To enhance the reproducibility of your results, we recommend that if applicable you deposit your laboratory protocols in protocols.io, where a protocol can be assigned its own identifier (DOI) such that it can be cited independently in the future. For instructions see: http://journals.plos.org/plosone/s/submission-guidelines#loc-laboratory-protocols

We look forward to receiving your revised manuscript.

Kind regards,

Jie Zhang

Academic Editor

PLOS ONE

Journal Requirements:

Reviewers' comments:

Reviewer's Responses to Questions

**Comments to the Author**

1. Is the manuscript technically sound, and do the data support the conclusions?

Reviewer #1: Yes

2. Has the statistical analysis been performed appropriately and rigorously? 

Reviewer #1: Yes

3. Have the authors made all data underlying the findings in their manuscript fully available?

Reviewer #1: Yes

4. Is the manuscript presented in an intelligible fashion and written in standard English?

Reviewer #1: Yes

5. Review Comments to the Author

Reviewer #1: In this paper, authors have trained ANN models on NIR spectra to estimate the parity status of wild mosquitoes based on four different datasets. Applying autoencoders in ANN is a way to develop smart chemometrics for rapid detections based on NIR technology. It is a somewhat valuable research. Nevertheless, some issues are also developing. I cordially raise some opinions for consideration and hope they can be of help for improving the paper.

Line 103: please clarify the way you control the killing temperature at -20oC?

Lines 127-129: how can you determine the division for training and test samples at the ratio of 75%/25%? I suppose 75% for training is over large because you don’t have a third sample set for model evaluation. Did you try other ratios?

Line 128: we can see that Minepa-ARA sample is much more than other samples, which will severely affect the prediction results for classification. Please demonstrate the methods for treating the problem of sample imbalance.

Line 280: please interpret the meaning of 2^2^(1851).

In Fig.2, how can you determine to use 10 feature nodes in the encoded feature space?

In Fig.4, can you explain why using logistic function for encoder’s activation but a linear function for decoder’s?

In Fig.8, you should identify the thresholds for each ROC curve.

6. PLOS authors have the option to publish the peer review history of their article (what does this mean?). If published, this will include your full peer review and any attached files.

Reviewer #1: No

---

## [Author Response · Author response to Decision Letter 0]

15 May 2020

May 7, 2020

Re: Resubmission of a Manuscript PONE-D-20-02482, An Autoencoder and Artificial Neural Network-based Method to Estimate Parity Status of Wild Mosquitoes from Near-infrared Spectra 

Jie Zhang

Academic Editor

PLOS ONE

Dear Editor: 

Thank you for the opportunity to revise our manuscript PONE-D-20-02482, An Autoencoder and Artificial Neural Network-based Method to Estimate Parity Status of Wild Mosquitoes from Near-infrared Spectra. We appreciate the careful review and constructive suggestions from you and the reviewer. It is our belief that the manuscript is substantially improved after making the suggested edits. 

Following this letter are the editor and reviewer comments with our responses, including how and where the text was modified. As suggested, we also have uploaded an unmarked version of the revised manuscript, along with a marked-up copy highlighting changes made to the original version. The revision has been developed in consultation with all co-authors, and each author has given approval to the final form of this revision.

Sincerely, 

Masabho P. Milali.

Reviewers’ comments:

General Comment

In this paper, authors have trained ANN models on NIR spectra to estimate the parity status of wild mosquitoes based on four different datasets. Applying autoencoders in ANN is a way to develop smart chemometrics for rapid detections based on NIR technology. It is a somewhat valuable research. Nevertheless, some issues are also developing. I cordially raise some opinions for consideration and hope they can be of help for improving the paper.

Author’s response:

Thank you.

Specific Comments 

Comment # 1:

Line 103: please clarify the way you control the killing temperature at -20o C?

Author’s response:

Prior to scanning, mosquitoes were killed by freezing them for 20 minutes in a freezer that is calibrated to -20o C. After freezing the mosquitoes were re-equilibrated to room temperature for 30 minutes. We have reflected this on lines 102 – 104. 

Comment # 2

Lines 127-129: how can you determine the division for training and test samples at the ratio of 75%/25%? I suppose 75% for training is over large because you don’t have a third sample set for model evaluation. Did you try other ratios?

Author’s response:

The ratio was picked based on the size of the data. We tried 70% / 30% data division and the results were consistent with the 75% / 25% split. 

Training and testing of the models presented in the manuscript was performed in MATLAB using 10-fold Monte Carlo cross validation. The technique generates estimates of variance and was described by Korjus et al (1). The library in MATLAB divides the 75% data into training, validation and testing making our 25% sample as the third sample set for model evaluation. 

We have modified the text on lines 130 – 133.

Comment # 3

Line 128: we can see that Minepa-ARA sample is much more than other samples, which will severely affect the prediction results for classification. Please demonstrate the methods for treating the problem of sample imbalance.

Author’s response:

To make sure that results from Minepa-ARA are not influenced by class imbalance, we repeated analysis while the number of mosquitoes in classes were matched (by randomly selecting mosquitoes from the class with large samples to match the number of mosquitoes from the class with few samples) and the results were similar. Furthermore, we computed and presented precision and recall (sensitivity) that are known to capture data imbalance (2). (Line 224 – 229). 

Comment # 4

Line 280: please interpret the meaning of 2^2^(1851).

Author’s response:

Let n be the number of binary features, then the number of hypotheses is 2^(2^n). To better understand why this is so, we need to split the problem into two parts. First, we need to determine how many unique instances there are. Second, we need to determine the number of possible sets of instance we can form. The answer to the second question is the size of the power set of X (2^|X|). The first is easy if we have binary features, just take 2 to the number of features we have. So, for one binary feature, we have two (2^1) unique instances; x = 0 or x = 1. The size of the power set of two instances is 2^2 or four. So we have four possible hypotheses for a single binary feature. The hypotheses are "never in the class", "when x == 0 in the class", "when x == 1 in the class", and "always in the class".

Comment # 5

In Fig.2, how can you determine to use 10 feature nodes in the encoded feature space?

Author’s response:

The number 10 was chosen to allow the salient features of the data to be identified. This is based on our knowledge of the data. 

Comment # 6

In Fig.4, can you explain why using logistic function for encoder’s activation but a linear function for decoder’s?

Author’s response:

We used logistic function for encoder part to capture the non-linear relationship between features. We used linear function for decoder to regenerate features in the domain similar to the original feature space.

Comment # 7

In Fig.8, you should identify the thresholds for each ROC curve.

Author’s response:

In all ROC curves, we calculated false positive rate and true positive rate using a 0.5 threshold. Reflected on lines 270 – 271, 325 – 326.

References

1. Korjus K, Hebart MN, Vicente R. An Efficient Data Partitioning to Improve Classification Performance While Keeping Parameters Interpretable. PloS One. 2016;11(8):e0161788.

2. Saito T, Rehmsmeier M. The Precision-recall Plot is More Informative than the ROC Plot When Evaluating Binary Classifiers on Imbalanced Datasets. PloS One. 2015;10(3):e0118432.

---

## [Decision Letter · Decision Letter 1]

29 May 2020

An Autoencoder and Artificial Neural Network-based Method to Estimate Parity Status of Wild Mosquitoes from Near-infrared Spectra

PONE-D-20-02482R1

Dear Dr. Milali,

We are pleased to inform you that your manuscript has been judged scientifically suitable for publication and will be formally accepted for publication once it complies with all outstanding technical requirements.

With kind regards,

Jie Zhang

Academic Editor

PLOS ONE

Additional Editor Comments (optional):

Reviewers' comments:

Reviewer's Responses to Questions

**Comments to the Author**

1. If the authors have adequately addressed your comments raised in a previous round of review and you feel that this manuscript is now acceptable for publication, you may indicate that here to bypass the “Comments to the Author” section, enter your conflict of interest statement in the “Confidential to Editor” section, and submit your "Accept" recommendation.

Reviewer #1: All comments have been addressed

2. Is the manuscript technically sound, and do the data support the conclusions?

Reviewer #1: Partly

3. Has the statistical analysis been performed appropriately and rigorously? 

Reviewer #1: Yes

4. Have the authors made all data underlying the findings in their manuscript fully available?

Reviewer #1: Yes

5. Is the manuscript presented in an intelligible fashion and written in standard English?

Reviewer #1: Yes

6. Review Comments to the Author

Reviewer #1: Authors have carefully modified the manuscript. I think the paper can be accepted at its current version.

7. PLOS authors have the option to publish the peer review history of their article (what does this mean?). If published, this will include your full peer review and any attached files.

Reviewer #1: No

---

## [Editor Report · Acceptance letter]

5 Jun 2020

PONE-D-20-02482R1 

An Autoencoder and Artificial Neural Network-based Method to Estimate Parity Status of Wild Mosquitoes from Near-infrared Spectra 

Dear Dr. Milali:

I'm pleased to inform you that your manuscript has been deemed suitable for publication in PLOS ONE. Congratulations! Your manuscript is now with our production department. 

Kind regards, 

on behalf of

Dr. Jie Zhang 

Academic Editor

PLOS ONE